## Research Article

CPTSD; COVID-19 worries; depression; loneliness; PTSD; trauma exposure

**Corresponding author:**
Lee Greenblatt-Kimron;
Email: leegr@ariel.ac.il

# Factors associated with ICD-11 posttraumatic stress disorder and complex posttraumatic stress disorder among older adults during the COVID-19 pandemic

Lee Greenblatt-Kimron[1] ![ORCID], Menachem Ben-Ezra[1], Maayan Shacham[2], Yaira Hamama-Raz[1] and Yuval Palgi[3]

[1]School of Social Work, Ariel University, Ariel, Israel; [2]Goldschleger School of Dental Medicine, Tel Aviv University, Tel Aviv, Israel and [3]Department of Gerontology, University of Haifa, Haifa, Israel

## Abstract

ICD-11 posttraumatic stress disorder (PTSD) and complex PTSD have been understudied in the older population. The study focused on the associations between traumatic exposure before the pandemic, COVID-19 worries, depression, and loneliness with current PTSD and CPTSD among older adults. A random sample of five hundred and twelve Israeli older adults ($M_{age}$ = 72.67 ± 3.81, range 68–87) was recruited using a Web-based survey company (*Ipanel, Israel*). Participants completed questionnaires of demographic details, self-rated health, COVID-19 worries, trauma exposure, depressive symptoms, level of loneliness, PTSD, and CPTSD. Univariate logistic regression revealed that trauma exposure, COVID-19 worries, depression, and loneliness were associated with PTSD. Multinomial regression revealed that only trauma exposure was associated with PTSD among older adults with PTSD compared with those not reaching the PTSD cutoff level. In the comparison between older adults suffering from CPTSD with those not reaching the PTSD cutoff level, being married, higher levels of trauma exposure, COVID-19 worries, depression, and loneliness were associated with a higher risk of CPTSD. Results suggest that specific factors may be significant psychological correlates of CPTSD symptoms among older adults during the COVID-19 pandemic. Identifying these factors could assist practitioners in tailoring more effective interventions.

## Impact statement

Posttraumatic stress disorder (PTSD) and ICD-11 complex PTSD (CPTSD) are considered under-researched disorders in the older population, particularly factors associated with CPTSD in old age. This study is unique as it aims to provide insight into the differences between older adults suffering from PTSD with those suffering from CPTSD. The study focused on factors associated with current PTSD and CPTSD among older adults exposed to a traumatic event before the COVID-19 pandemic. This study differentiated between three groups of older adults at the end of the fifth wave of the COVID-19 pandemic in Israel: those with no diagnosis of PTSD, those who reached the PTSD cutoff, and those who exhibited CPTSD. Participants with PTSD reported higher levels of trauma exposure, COVID-19 worries, depression, and loneliness relative to those not reaching the PTSD cutoff level. Moreover, participants with CPTSD displayed higher levels of loneliness, COVID-19 worries, exposure to trauma, and depression relative to those not reaching the PTSD cutoff level. The study underscores that older adults suffering from trauma are at risk for adverse outcomes in the face of global disasters, such as the COVID-19 pandemic. Furthermore, there is a significant risk for those suffering from CPTSD, distinguishing them as particularly vulnerable to the adverse effects of global disasters. It seems that complex trauma is significantly more related to mental health factors among older adults, which has clinical implications for practitioners. The study highlights the need for further research in the field of CPTSD among older adults to enhance suitable preventive interventions for the adverse mental outcomes of older adults faced with global disasters, with extra consideration for extended treatment protocols for those suffering from CPTSD.

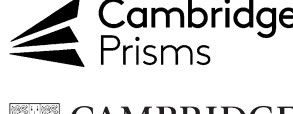

## Introduction

Older adults have been particularly affected by the adverse outcomes of the COVID-19 pandemic, with many being infected by the virus (Niu et al., 2020). Consequently, studies focusing on older adults throughout the pandemic have reported declines in mental health (Shrira et al., 2020)

including death anxiety (Ring et al., 2022), peritraumatic distress (Greenblatt-Kimron et al., 2021), and posttraumatic stress disorder (PTSD) (Palgi et al., 2021). Moreover, older adults suffering from PTSD depression comorbidity before the outbreak of COVID-19 reported the highest risk of feeling depressed, anxious, and lonely relative to those with no symptoms before the pandemic (Cohn-Schwartz et al., 2022). Furthermore, researchers have described COVID-19 as a persistent continuing traumatic stress, with a probability of being encountered as severe (Kira et al., 2021).

The ICD-11 broadened the knowledge of traumatic stress disorders with the new disorder of complex PTSD (CPTSD) (World Health Organization, 2018). CPTSD, linked with extended, recurrent, interpersonal, and unpreventable trauma exposure (Karatzias et al., 2017), comprises three PTSD clusters of re-experiencing the trauma, avoidance of traumatic reminders, and a continual sense of threat and also includes three clusters reflecting disturbances in self-organization (DSO), namely affective dysregulation, negative self-concept, and disturbances in relationships (World Health Organization, 2018). Researchers examined PTSD among older adults during the COVID-19 pandemic (Palgi et al., 2021); nonetheless, PTSD and CPTSD are considered under-researched disorders in the older population (World Health Organization, 2017), particularly factors associated with CPTSD in old age. This study is unique as it specifically aims at examining older adults while providing insight into the differences between older adults suffering from PTSD with those also suffering from CPTSD. We assume that under conditions of prolonged COVID-19 stressors, those with higher previous exposure to traumatic events will exhibit higher vulnerability to PTSD and CPTSD. In particular, the study examined factors associated with current PTSD and CPTSD among older adults at the end of the fifth wave of COVID-19 in Israel. These factors, discussed briefly as follows, include previous trauma exposure, COVID-19 worries, depression, and loneliness.

### Prior trauma exposure and posttraumatic reactions

Previous data indicate that adults aged between 60 and 70 years have about 3.5 and 2.5 times less chance of reaching the diagnostic requirements for PTSD and CPTSD, respectively, than those younger than 60 years (Fox et al., 2022). Nevertheless, data from samples from four different countries show that PTSD rates were generally lower among older adults, while inconsistent findings are found with regard to CPTSD (McGinty et al., 2021). Significantly, previous studies show a link between PTSD from previous trauma exposure with a heightened risk for PTSD succeeding a subsequent traumatic event (Breslau et al., 2008; Kessler et al., 2018). Moreover, prior trauma exposure may amplify vulnerability during the COVID-19 pandemic and correlate with successive posttraumatic reactions (Lahav, 2020). Indeed, in a study among the general population in Israel, anxiety, depression, and peritraumatic stress symptoms linked with COVID-19 were higher among those with pre-exposure to traumatic events before the COVID-19 pandemic relative to those without trauma exposure, invariant of demographic characteristics, and pandemic stressors (Lahav, 2020). Furthermore, in the first study to examine COVID-19 vaccine-related stressors in the context of ICD-11 PTSD symptoms among older adults exposed to a traumatic event before the COVID-19 outbreak, results showed an exacerbation of PTSD symptoms due to depression, ageism, and COVID-19 vaccine-related factors (Palgi et al., 2021). On this basis, it is assumed that older adults with trauma exposure before the COVID-19 pandemic will exhibit higher levels

of PTSD and CPTSD relative to older adults without trauma exposure.

### COVID-19 worries and posttraumatic reactions

Researchers highlighted the adverse effects of COVID-19 worries on older adults' mental health (Greenblatt-Kimron et al., 2021). For example, a relationship was reported between perceived susceptibility to COVID-19 with worry, fear, and stress as an outcome of the pandemic (Inbar and Shinan-Altman, 2021). Other researchers found correlations between COVID-19 health worries with death anxiety (Ring et al., 2022) and peritraumatic distress (Greenblatt-Kimron et al., 2021). In this light, it is assumed that COVID-19 worries would be associated with current PTSD and CPTSD among older adults.

### Depressive symptoms and posttraumatic reactions

It is well documented that untreated depressive symptoms may harm older adults' physical and mental health (Agustini et al., 2020). While sparse research examined ICD-11 PTSD comorbidity among older adults (Fox et al., 2020), a recent study identified comorbidity with major depressive disorder (Fox et al., 2020). With respect to the COVID-19 pandemic, numerous studies revealed depressive symptoms among older adults (Palgi et al., 2020; Shrira et al., 2020) including those who had received the COVID-19 vaccination (Greenblatt-Kimron et al., 2022). Despite vaccination programs being visualized as the "light at the end of the tunnel" (Hoffman et al., 2022), it was reported in a sample of community-dwelling older adults that 23.8% reached the clinical level of ICD-11 PTSD despite receiving a second dose of the COVID-19 vaccine, with depressive symptoms identified as one of the main factors linked with PTSD (Palgi et al., 2021). Therefore, this study aimed to broaden the understanding of the relationship between depressive symptoms with traumatic reactions among older adults by including CPTSD in the examination of the adverse mental effects of COVID-19 on older adults who had experienced previous trauma before the outbreak of the pandemic.

### Loneliness and posttraumatic reactions

Additionally, the COVID-19 pandemic was described as the "loneliness pandemic" (Palgi et al., 2020). Loneliness, defined as the personal perception of unfulfilled intimate, emotional, and social needs (Dykstra, 2009), is known for its detrimental mental and physical health outcomes in the older population (Courtin and Knapp, 2015). Previous studies reported associations between loneliness and PTSD (Shevlin et al., 2015; Itzhaky et al., 2017), with loneliness playing a significant role in contributing to or maintaining CPTSD symptoms (Murphy et al., 2021; Fox et al., 2022). Nevertheless, research regarding the association between loneliness and CPTSD is scarce (Fox et al., 2022), particularly in the older population. Among the few existing studies, a correlation was reported between longitudinal changes in social and emotional loneliness with longitudinal changes in PTSD symptoms (Fox et al., 2021). Likewise, a longitudinal study of Israeli prisoners showed a link between those exhibiting CPTSD symptoms and an overall sense of loneliness later in life compared with those exhibiting PTSD symptoms or were without symptoms (Zerach et al., 2019). Concerning the COVID-19 pandemic, loneliness was associated with higher levels of anxiety, depression, and peritraumatic distress symptoms among older adults, particularly among

those who perceived themselves as older (Shrira et al., 2020). Hence, the present study sought to also examine the relationship between loneliness and current PTSD and CPTSD among older adults during the fifth COVID-19 wave in Israel, with an emphasis on those suffering from CPTSD.

### The current study

Previous data indicated links between trauma exposure, COVID-19 worries, depression, and loneliness with PTSD symptoms. Aimed at extending the knowledge on ICD-11 PTSD and CPTSD among older adults, highlighted as under-researched disorders in the older population (World Health Organization, 2017), this study examined the associations between these factors with PTSD and CPTSD among older adults during the COVID-19 pandemic. We hypothesized that previous trauma exposure would be positively associated with current PTSD. The second hypothesis, based on data highlighting the adverse mental outcomes of COVID-19 worries among older adults (Inbar and Shinan-Altman, 2021), maintained that a positive relationship would be found between COVID-19 worries and current PTSD. The third hypothesis was assumed to confirm the reputed association previously found between depression with current PTSD among vaccinated older adults during the COVID-19 pandemic (Palgi et al., 2021). The fourth hypothesis was elicited from the suggestion that the COVID-19 pandemic is a "loneliness pandemic" (Palgi et al., 2020), hypothesizing a relationship between loneliness and current PTSD. Finally, we hypothesized that the associations with the study variables (previous trauma exposure, higher COVID-19 worries, depression, and loneliness) would be stronger among those exhibiting CPTSD.

### Materials and methods

Using a Web-based survey company (*Ipanel, Israel*), data were collected at the end of the fifth wave in Israel during May 2022. During that time, over 95% of the older population had been vaccinated in Israel. Therefore, although we did not examine COVID-19 exposure, it would be fair to assume that almost all participants had received the vaccine. After omitting those who reported that they had not experienced any traumatic event, participants included 512 older Israeli adults aged 67 and older (average age = 72.67 ± 3.81, range 68–87). The inclusion criteria for the study included being fluent in Hebrew and being over the age of 67 (retirement age for males in Israel). The study was approved by the Ariel University IRB Committee. Participants signed an electronic informed consent form and completed self-report questionnaires. About half of the respondents were female (*n* = 257, 50.2%), 71.5% were in a committed relationship (*n* = 366), and 81.8% had tertiary education (*n* = 419) (see Table 1).

### Measurements

*Background characteristics* included age, gender, marital status (1 = not married, 2 = married/living as cohabitant), and years of education. Physical health was reported using the self-rated health item "In general, how do you rate your health?" on a scale ranging from 1 (*not good at all*) to 5 (*very good*) (Idler and Benyamini, 1997).

*Exposure to previous traumatic events* was measured by the Potentially Traumatic Events Inventory adapted from Shmotkin and Litwin (2009). Participants were instructed to indicate if they experienced a particular event (yes/no) regarding 13 events (e.g. *You were injured in a war or a terrorist attack*). On average, participants reported that they were exposed to 3.50 (*SD* = 2.06) traumatic events. The most frequent events reported were the *sudden death of a close person* (78.7%) and *life-threatening event of a close a person* (e.g. *illness, injury, and physical impairment*) (53.1%). The sum of "*yes*" answers was used to determine the level of exposure.

*COVID-19 worries* were assessed by the Brief Coronavirus Threat Scale (Chiacchia et al., 2022). Participants were instructed to report how they felt about each of five statements (e.g. *How much*

**Table 1.** Descriptive statistics for the study variables

| | *M*/% | *SD* | 1 | 2 | 3 | 4 | 5 | 6 | 7 | 8 | 9 | 10 |
|---|---|---|---|---|---|---|---|---|---|---|---|---|
| 1. Age | 72.67 | 3.81 | – | | | | | | | | | |
| 2. Gender[a] | 50.2% | – | −0.19*** | – | | | | | | | | |
| 3. Marital status[b] | 71.5% | – | −0.04 | −0.27*** | – | | | | | | | |
| 4. Education[c] | 4.42 | 1.22 | −0.04 | 0.02 | 0.09* | – | | | | | | |
| 5. Self-rated health[d] | 3.33 | 0.95 | −0.08 | −0.01 | 0.02 | −0.13** | – | | | | | |
| 6. Previous traumatic exposures | 3.50 | 2.06 | −0.01 | −0.25*** | −0.03 | 0.04 | −0.10* | – | | | | |
| 7. COVID-19 worries | 1.20 | 0.88 | −0.02 | 0.13** | −0.11* | −0.03 | −0.32*** | 0.15*** | – | | | |
| 8. Depressive symptoms | 4.94 | 4.49 | 0.01 | 0.18*** | −0.17*** | −0.08 | −0.37*** | 0.13** | 0.47*** | – | | |
| 9. Loneliness | 15.76 | 4.69 | 0.02 | 0.09 | −0.11* | 0.07 | −0.22*** | 0.13** | 0.37*** | 0.53*** | – | |
| 10. PTSD[e] | 7.4% | – | −0.04 | 0.06 | −0.07 | −0.05 | −0.13** | 0.25*** | 0.38*** | 0.36*** | 0.25*** | – |

*Note: Total N = 512.*
[a]woman.
[b]currently married or living with a partner.
[c]ranging on a scale from 0 = no formal education to 5 = high academic education.
[d]1 = not good at all to 5 = very good.
[e]clinical level of PTSD.
*p < 0.05;
**p < 0.01;
***p < 0.001.

*do you feel threatened about Coronavirus?*) on a five-point Likert scale from 1 (*not at all*) to 5 (*extremely/a great deal*). A total score is computed by calculating the mean response to five items. Cronbach's α in the current study was α = 0.920.

*Depressive symptoms* were assessed by the 9-item scale (PHQ-9, Kroenke et al., 2001). Participants rated how often they felt depressive symptoms in the last month, on a four-point Likert scale from 0 (*not at all*) to 3 (*nearly every day*). Scores were computed by summing the items, and ranged from 0 to 24, with higher scores reflecting higher levels of depressive symptoms. Cronbach's α in this study was α = 0.844.

*Loneliness* was assessed by the short version of the Revised UCLA Loneliness Scale (UCLA-8) (Hays and DiMatteo, 1987). This scale includes eight items (e.g. *I lack companionship*) that measure the participants' level of subjective distress due to lack of relationships, on a scale ranging from 1 (*never*) to 4 (*very often*). Scores were computed by the sum of the item scores and ranged from 8 to 32, with higher scores indicating greater loneliness. Cronbach's α in this study was α = 0.810.

*Probable Posttraumatic Stress Disorder (PTSD) and Complex Posttraumatic Stress Disorder (CPTSD)* were assessed by the International Trauma Questionnaire (ITQ; Cloitre et al., 2018). *Probable PTSD* was measured by six items and was determined when participants reported a high or very high level of suffering (≥2) on a five-point Likert scale ranging from 0 (*not at all*) to 4 (*extremely*) in at least one item of each of the three clusters of re-experiencing, avoidance, and sense of threat. In addition, they had to report that these symptoms caused significant functional impairment (≥2) as reported by three items of important areas of life. *Probable CPTSD* was determined when participants fit the probable PTSD criteria and also fit the disturbance of self-organization (DSO) criteria. The DSO was assessed with six items and was determined when participants reported a high or very high level of suffering (≥2) in at least one item of each of the three clusters, affective dysregulation, negative self-concept, and disturbed relationships. In addition, to determine probable CPTSD they had to report that these symptoms cause significant functional impairment (≥2) as reported by three items of important areas of life. For convenience purposes, we refer to those reaching clinical levels of PTSD or CPTSD, simply as PTSD or CPTSD. Cronbach's α for the 12 items (PTSD+CPTSD) in this study was α = 0.897.

## Data analysis

First, correlations were examined to establish preliminary associations between the study variables. Then, univariate logistic regressions were conducted to examine the association between COVID-19 worries, depression, and loneliness in older adults and PTSD (and CPTSD, together) cutoffs compared with those not reaching cutoff levels. Next, multinomial logistic regressions were conducted to examine the association between COVID-19 worries, depression, and loneliness in older adults reporting PTSD only compared with those with no diagnosis and CPTSD compared with those with no diagnosis. Using SPSS version 27.0, we calculated the odds ratio (OR) with 95% confidence intervals to quantify the strength of the associations.

## Results

In our sample, 38 (7.4%) participants reached the clinical PTSD criteria level. Of these, 21 (4.1%) reported a clinical level of PTSD, while the other 17 (3.3%) also reported CPTSD. Participants with PTSD and CPTSD, compared with those without PTSD, showed higher levels of COVID-19 worries, higher levels of trauma exposure, and higher levels of depression ($M$ = 2.20/2.59 vs. 1.11, $F$ = 43.31, $p < 0.001$, $\eta^2$ = 0.15; $M$ = 4.81/6.00 vs. 3.35, $F$ = 19.23, $p < 0.001$, $\eta^2$ = 0.07; $M$ = 8.29/13.47 vs. 4.48, $F$ = 45.90, $p < 0.001$, $\eta^2$ = 0.15, respectively). Moreover, only participants with CPTSD showed higher levels of loneliness compared with those without PTSD ($M$ = 17.43/23.06 vs. 15.42, $F$ = 25.32, $p < 0.001$, $\eta^2$ = 0.09).

Next, we conducted a univariate logistic regression to examine differences between participants with PTSD (and CPTSD, together) compared with those who do not reach a cutoff level of PTSD. Results showed that higher levels of COVID-19 worries, trauma exposure, depression, and loneliness increased the risk of having PTSD (see Table 2). Finally, we examined whether there are differences in these associations between participants with PTSD and those who also suffered from CPTSD. Therefore, we conducted a multinomial logistic regression. Analysis showed that in the comparison between those suffering from PTSD relative to those who did not reach the PTSD cutoff level, only COVID-19 worries and trauma exposure were significantly associated with PTSD ($OR$ = 2.97; 95% CI: 1.69–5.21, $p < 0.001$; $OR$ = 1.30, 95% CI: 1.05–1.62, $p < 0.05$, respectively).

However, in comparison with those who suffered from CPTSD with those who do not reach the PTSD cutoff level, those who were married ($OR$ = 5.73, 95% CI: 1.10–29.67), and who had higher levels of COVID-19 worries ($OR$ = 5.48, 95% CI: 1.88–16.02), exposure to trauma ($OR$ = 1.83, 95% CI: 1.34–2.50), depression ($OR$ = 1.25, 95% CI: 1.09–1.44), and loneliness ($OR$ = 1.29, 95% CI: 1.08–1.53) also showed higher levels of CPTSD. Although no sex differences were found between men and women in the logistic regression, we conducted an additional t-test analysis to examine whether there were differences between males and females with the sum of PTSD and CPTSD symptoms. The results of the t-test revealed a significant difference between the sexes, with females ($M$ = 9.03, $SD$ = 6.93 vs. $M$ = 7.29, SD = 6.94, $t(510)$ = −264, $p < 0.01$, $\eta^2$ = −0.23; $M$ = 4.33 $SD$ 4.22 vs. $M$ = 3.49 $SD$ 3.76; $t$ (510) = −2.37, $p < 0.05$, $\eta^2$ = −0.23, respectively) showing higher PTSD and CPTSD symptoms than males. The effect size was small.

## Discussion

The current study assessed factors associated with current PTSD and CPTSD among older adults exposed to a traumatic event before the COVID-19 pandemic. The study was based on the ICD-11 definitions of PTSD and CPTSD, which are under-examined disorders in the context of older adults (World Health Organization, 2017). In our sample, the prevalence of PTSD was 7.4%, of which 4.1% reported PTSD, while the other 3.3% also reported CPTSD. These findings coincide with those of a representative sample of older adults in the United States, where 6.1% met the criteria of ICD-11 PTSD (Fox et al., 2020). Compared to studies during the COVID-19 pandemic in which the percentages were higher (23.8%) (Palgi et al., 2021), it seems that the discovery of the vaccine, the decrease in morbidity, and the return to routine restored the level of trauma to its normal level.

The current study differentiated between three groups of older adults at the end of the fifth wave of the COVID-19 pandemic in Israel: those with no diagnosis of PTSD, those who reached the PTSD cutoff, and those who exhibited CPTSD. As hypothesized, participants with PTSD reported higher levels of trauma exposure, COVID-19 worries, depression, and loneliness. Moreover,

**Table 2.** Multinomial and univariate logistic regression analyses

| Variable | Univariate logistic regressions, Likelihood of diagnosis relative to No diagnosis | Multinomial logistic regressions, Likelihood of diagnosis relative to No diagnosis | |
| --- | --- | --- | --- |
| | PTSD and CPTSD cutoff (474 vs. 38) OR (95% CI) | PTSD cutoff (474 vs. 21) OR (95% CI) | CPTSD cutoff (474 vs. 17) OR (95% CI) |
| Age | 0.969 (0.872–1.076) | 0.871 (0.743–1.020) | 1.109 (0.947–1.300) |
| Gender[a] | 1.411 (0.567–3.511) | 1.093 (0.373–3.204) | 2.584 (0.560–11.917) |
| Marital status[b] | 1.624 (0.625–4.217) | 0.928 (0.308–2.796) | 5.725 (1.105–29.672)* |
| Education[c] | 0.806 (0.565-1.152) | 0.857 (0.570–1.291) | 0.677 (0.353–1.300) |
| Self-rated health[d] | 1.175 (0.739–1.868) | 1.032 (0.599–1.778) | 1.603 (0.708–3.630) |
| Previous traumatic exposures | 1.422 (1.189–1.701)*** | 1.303 (1.049–1.619)* | 1.831 (1.341–2.498)*** |
| COVID-19 worries | 3.311 (1.992–5.501)*** | 2.969 (1.694–5.205)*** | 5.482 (1.876–16.017)** |
| Depressive symptoms | 1.128 (1.033–1.232)* | 1.065 (0.958–1.183) | 1.251 (1.085–1.443)** |
| Loneliness | 1.076 (0.981–1.180)* | 1.003 (0.897–1.121) | 1.286 (1.080–1.531)** |

*Note: Total N = 512; Nagelkerke $R^2$ = 0.47 and 0.44, respectively.*
[a]Gender 1 = male, 2 = female.
[b]Marital status 1 = not married, 2 = currently married or living with a partner.
[c]= Five education levels from 1 = preprimary education to 5 = tertiary education.
[d]1 = not good at all to 5 = very good.
*$p < 0.05$;
**$p < 0.01$;
***$p < 0.001$.

participants with CPTSD displayed higher levels of loneliness than those with PTSD and higher levels of COVID-19 worries, exposure to trauma, depression, and loneliness compared with those without PTSD. These results will be further discussed.

In line with the first hypothesis, the results of the univariate logistic regression indicated a link between higher levels of trauma exposure with general PTSD. These results support previous studies showing a relationship between trauma exposure and PTSD and an increased risk of heightened PTSD after an additional traumatic event (Breslau et al., 2008; Kessler et al., 2018). Moreover, as suggested by Lahav (2020), trauma survivors, particularly those exposed to continuous traumatic stress, appear to be at heightened risk for psychological distress linked with COVID-19. In addition, our findings coincide with results showing a link between peritraumatic stress symptoms due to COVID-19 (Greenblatt-Kimron et al., 2021) and elevated levels of PTSD among older adults who had experienced a traumatic event before the outbreak of the pandemic (Palgi et al., 2021).

Consistent with the second study hypothesis, a positive association was found between COVID-19 worries and PTSD. This finding may be understood by the adverse physical and mental outcomes of the COVID-19 pandemic on older adults (Niu et al., 2020; Shrira et al., 2020) and supports a study in Israel, showing a link between COVID-19 worries and peritraumatic distress among older adults at the outbreak of the pandemic (Greenblatt-Kimron et al., 2021).

Supporting the third hypothesis, the current data showed a relationship between higher levels of depression with PTSD among older adults during the COVID-19 pandemic. This is in line with a previous study revealing an association between depressive symptoms and clinical levels of ICD-11 PTSD among older adults who had received the COVID-19 vaccine (Palgi et al., 2021). Likewise, the current finding affirms the conclusion that cumulative adversity increases the prospect of depressive symptoms in later life (Shmotkin and Litwin, 2009) and is in consonance with the findings of ICD-11 PTSD comorbidity with major depressive disorder in a sample of community-dwelling older adults (Fox et al., 2020).

In accordance with the fourth hypothesis, during the COVID-19 pandemic, described as the "loneliness pandemic" (Palgi et al., 2020), older adults showing PTSD reported more loneliness. This finding is consistent with previous data showing a link between loneliness and higher levels of psychiatric symptoms, specifically anxiety, depression, and peritraumatic distress symptoms among older adults (Shrira et al., 2020).

In the univariate analysis, most variables were associated with PTSD. Nonetheless, the multinomial analysis showed that most of the effect actually comes from CPTSD. This finding strengthens the notion that CPTSD is associated with numerous forms of traumatic experiences, and while PTSD may be triggered by a single event, CPTSD is generally the outcome of exposure to prolonged or cumulative trauma (Karatzias et al., 2017). Furthermore, the current results support findings showing significant differences in the number of traumas experienced by a sample of Israeli adults, with the lowest number reported by those with no probable diagnosis and the highest number by those with probable CPTSD (Greenblatt-Kimron et al., 2023). Additionally, results of the current study showing higher levels of COVID-19 worries and depressive symptoms among those exhibiting CPTSD support previous results, indicating that those with probable CPTSD are the most affected and show a higher level of other comorbidities than those with PTSD, such as symptoms of depression and anxiety (Hyland et al., 2020).

Moreover, our findings indicate that those with CPTSD reported higher levels of loneliness than those with PTSD. These results have significance in trauma studies among older adults. Previous research has underscored the loneliness–PTSD link (Shevlin et al., 2015; Itzhaky et al., 2017) and the role of loneliness in the contribution to and maintenance of CPTSD symptoms (Murphy et al., 2021; Fox et al., 2022). Nevertheless, insufficient research based on ICD-11 PTSD and CPTSD examined this relationship in older populations (Fox et al., 2022). In this light, the current results add to those of a recent study among a nationally representative sample of US community-dwelling older adults

between the ages of 60 and 70, showing a relationship between emotional loneliness with symptoms of ICD-11 PTSD and DSO (CPTSD) symptoms, while social loneliness was only associated with DSO symptoms (Fox et al., 2022). The current results also support findings among treatment-seeking veterans, indicating a correlation between feelings of social isolation and loneliness with an anticipated CPTSD diagnosis (Murphy et al., 2021).

The present data indicated no significant differences between those with PTSD compared with those without PTSD and those with CPTSD compared with those without PTSD in terms of age or education. However, t-test analysis revealed a significant difference between the sexes, with females reporting higher PTSD and CPTSD symptoms than males. Our finding supports previous findings, showing higher rates of PTSD among women in the DSM criteria of PTSD (e.g. Olff, 2017) and in the PTSD diagnostic criteria in the ICD-11 (e.g. McGinty et al., 2021). Nevertheless, in a recent study of four representative samples from the United Kingdom, the Republic of Ireland, the United States, and Israel, women met the criteria for CPTSD more than men only in the US sample (McGinty et al., 2021). The researchers recommend further studies on CPTSD, as data reveal inconsistent findings of sex and age differences, and perhaps even an interaction between the two (McGinty et al., 2021).

Additionally, our results suggest that in comparison with those suffering from CPTSD with those who do not reach the PTSD cutoff level, married older adults reported higher levels of CPTSD. This finding was unexpected as previous research focusing on the benefits of marriage reported links between marriage with better mental health, including higher life satisfaction (Fu and Noguchi, 2016) and lower depression and anxiety (Purba and Fitriana, 2019). Nonetheless, the current findings may be understood in light of the pandemic era, whereby researchers questioned whether being married served as a buffer to the stresses and policies foisted by COVID-19 (Purba et al., 2021), as fear, worry, and fundamental existential survival became paramount, which may result in the loss of the essential support provided by marital bonding (Maiti et al., 2020). Likewise, previous studies examining ICD-11 CPTSD showed mixed results with regard to relationship status, with some researchers reporting a link between CPTSD and living alone or not being married (Karatzias et al., 2017), while others found no differences in relationship status and CPTSD (Greenblatt-Kimron et al., 2023). Nonetheless, the current findings need to be interpreted with caution as most of the sample was married.

Altogether, the current study underscores that those suffering from trauma are at risk for adverse outcomes in the face of continuous stress, such as the COVID-19 pandemic. This is accentuated by the current study taking place during the fifth wave of the COVID-19 pandemic, with an emphasis on examining COVID-19 worries among older adults with previous trauma exposure. In this light, the result highlights the vulnerability of older adults with previous trauma in the face of global disasters. Furthermore, the current study underscores that there is a significant risk for those suffering from CPTSD during global disasters, such as the COVID-19 pandemic, distinguishing them as particularly vulnerable.

The present study has limitations. First, due to the cross-sectional nature of the study causality cannot be inferred. Second, the data were collected through a Web-based online survey company and may be biased toward older adults with high education and technological skills. Third, the study was based on self-report questionnaires, which may limit the ability to generalize the findings. Fourth, a major limitation of the study is that the PTSD and CPTSD groups are very small; therefore, caution should be given in generalizing the results. Future studies examining PTSD and

CPTSD among the older population during global crises are thus recommended. Fifth, we did not measure participants' exposure level to COVID-19. Finally, there are no data regarding response rates, limiting the finding's generalizability.

## Conclusions

Despite the above limitations, the present study is among the first to examine ICD-11 PTSD and CPSTD among older adults, particularly during the COVID-19 pandemic. These findings have theoretical and practical implications for the risk factors associated with PTSD and CPTSD in the older population during the COVID-19 pandemic, as well as other natural disasters. Theoretically, the findings enhance previous results by revealing the relationships between increased exposure to trauma, COVID-19 worries, depression, and loneliness with PTSD and CPTSD. The findings underscore that those suffering from CPTSD may be at risk and vulnerable to the adverse effects of the pandemic. Nevertheless, due to the study's cross-sectional nature, future longitudinal studies should examine directionality, namely whether those with CPTSD suffer more from the pandemic, or whether the pandemic exacerbates their conditions. Overall, the current study adds to the existing knowledge concerning the adverse outcomes of pandemics on the mental health of older adults by underscoring the association between COVID-19 worries with higher levels of PTSD and CPTSD among those exposed to a traumatic event before the outbreak of the COVID-19 pandemic. As the development of suitable interventions for the treatment of PTSD, particularly for CPTSD, is necessary for older adults (Fox et al., 2022), specifically in the context of ongoing global natural and man-made disasters, future studies are recommended to examine the directionality of this relationship and its contribution to mental health among older adults during global crises and catastrophes.

On a practical level, the findings offer preliminary support for suitable interventions for older adults during the COVID-19 pandemic, aimed at reducing health worries. In particular, the findings add to previous data (Greenblatt-Kimron et al., 2021), which underscore the importance of identifying health worries among older adults, and more so among those with prior trauma exposure. The results support the suggestion that individuals exposed to trauma are at a heightened risk of the harmful effects of persistent stress, such as the COVID-19 pandemic (Lahav, 2020). Significantly, the study underscores that older adults suffering from CPTSD are particularly vulnerable to the adverse effects of global disasters. In this light, preventive interventions for adverse mental outcomes should be tailored for older adults, with extra consideration for extended treatment protocols specific for CPTSD, due to the higher number and types of symptoms and the increased functional impairment (Karatzias et al., 2017).

**Open peer review.** To view the open peer review materials for this article, please visit http://doi.org/10.1017/gmh.2023.42.

**Data availability statement.** Not available to ensure participants' confidentiality.

**Author contribution.** Lee Greenblatt-Kimron and Yuval Palgi provided substantial contributions to the conception or design of the work; acquired, analyzed, and interpreted the data. Lee Greenblatt-Kimron drafted the data. Lee Greenblatt-Kimron, Menachem Ben-Ezra, Maayan Shacham, Yaira Hamama-Raz, and Yuval Palgi[3] revised the manuscript critically for important intellectual content. Lee Greenblatt-Kimron, Menachem Ben-Ezra, Maayan Shacham, Yaira Hamama-Raz, and Yuval Palgi[3] approved the final version of the article to be

published. Lee Greenblatt-Kimron, Menachem Ben-Ezra, Maayan Shacham, Yaira Hamama-Raz, and Yuval Palgi[3] agreed to be accountable for all aspects of the work in ensuring that questions related to the accuracy or integrity of any part of the work are appropriately investigated and resolved.

**Financial support.** This study was supported by an internal research grant from Ariel University, given to Lee Greenblatt-Kimron.

**Competing statement.** The authors declare none.

**Ethics standard.** This study received ethical approval from the Ariel University IRB Committee, Number: AU-SOC-LG-20220424.

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
