## [Reviewer Report]

April 9, 2023

Professor Gary Belkin

Editor 

Global Mental Health 

Dear Professor Belkin,

We herewith submit our paper entitled “Factors associated with ICD-11 posttraumatic stress disorder (PTSD) and complex PTSD among older adults during the COVID-19 pandemic" for consideration for publication in your journal, Global Mental Health.

We declare that the contents of this manuscript have not been copyrighted or published previously and are not under consideration for publication elsewhere.

The contents of this manuscript will not be copyrighted, submitted or published elsewhere while acceptance by the journal is under consideration.

Sincerely,

Lee Greenblatt-Kimron, Menachem Ben-Ezra, Maayan Shacham, Yaira Hamama-Raz & Yuval Palgi

Corresponding Author: 

Lee Greenblatt-Kimron 

School of Social Work, Ariel University, Ariel, Israel University.

E-mail: leegr@ariel.ac.il

---

## [Reviewer Report]

The current study is important as few works looked at CPTSD among older adults. I read the manuscript with much interest, and I think that overall, the study is strong and contributes to the literature. However, I had several concerns regarding the description of findings and the conclusions.

Abstract & main text: Please provide more details about the sample: was it random, stratified? Correct multinominal to Multinomial regression.

Abstract: The following sentence is unclear: “trauma exposure was associated with PTSD among older adults with PTSD compared to those no reaching the PTSD cutoff”; did you mean trauma exposure was associated with PTSD? I guess this analysis compared those with PTSD to those without, right? The same goes to the next sentence: I think you wanted to say that these variables were associated with a higher risk of CPTSD.

Impact statement: The findings are not described clearly in the impact statement as well. For example: “Participants with PTSD reported higher levels of trauma exposure, COVID-19 worries, depression and loneliness (relative to whom?). The next sentence is also unclear: Did those with CPTSD have higher levels of COVID-19 worries, exposure to trauma and depression relative to those with PTSD? Moreover, the sentence referring to people in the second half of life is not accurate. The sample included those age 68 and above.

Main text: Authors should refer to previous works on older adults with PTSD during the pandemic: Cohn-Schwartz, E., Hoffman, Y., & Shrira, A. (2022). The effect of pre-pandemic PTSD and depression symptoms on mental distress among older adults during COVID-19. Journal of psychiatric research, 151, 633-637.

In p. 5: “on this basis, it is assumed that older adults with trauma exposure prior to covid-19 pandemic will exhibit higher levels of PTSD and CPTSD in comparison to older adults without PTSD and CPTSD.” This sentence makes no sense. I guess you wanted to say: older adults with trauma exposure prior to covid-19 pandemic will exhibit higher levels of PTSD and CPTSD in comparison to older adults without trauma exposure.

P 9: “we hypothesized that previous trauma exposure would be positively associated with current PTSD among participants who had trauma exposure prior to the pandemic”. Again, this phrasing makes no sense… Moreover, the sentence: “these associations would be stronger among those exhibiting CPTSD” makes no sense either. The previous hypotheses referred to associations between various factors and PTSD.

The text needs editing. There are many examples… in p 6: “light of the end of the tunnel”; p. 14: “the role of loneliness in the contribution to and maintained of CPTSD…”; p. 17: “and more so among those who prior trauma exposure”

It would be important to report covid-19 exposure level for the sample. Moreover, in these ages many individuals report more than one traumatic event. What were the instructions regarding the PTSD symptoms when respondents reported more than one event: do we know to which event their referred to when they reported several events?

Please report frequencies for the traumatic events reported.

The PTSD and CPTSD groups are very small – this is a major limitation that should be acknowledged and discussed.

When presenting the f tests please provide effect size statistics.

P 11: “the results showed that trauma exposure… had a higher association with PTSD”. The phrasing is unclear. Higher association with PTSD compared to what? Perhaps use the following: “higher levels of trauma exposure… increased the risk of having PTSD” etc. I suggest the authors revise their report of the logistic and multinomial regressions accordingly.

p. 12: I don’t think the prevalence of PTSD/CPTSD in the current study is similar to the prevalence in Hyland et al. 2020 – in the Hyland study the prevalence seems to be twice as higher (11.6 to 9 vs 5).

p. 12: the authors compare the prevalence in their study (5%) to the prevalence of PTSD found in Palgi et al 2021 (23.8%). The difference is indeed striking (especially as both studies used similar samples and measures) and I would emphasize the difference by referring to the specific % found in each study.

p. 13: “the current results strengthen the suggestion that prior trauma exposure may augment vulnerability during the covid-19 pandemic” or p. 17: “the results support the suggestion that individuals exposed to trauma are at a heightened risk of harmful effects due to persistent stress, such as the covid-19 pandemic” – this cannot be the conclusion following the current study as all respondents reported to have been experiencing at least one traumatic event before the pandemic.

---

## [Reviewer Report]

This is an interesting study that tests the hypothesis that stress-related disorders are important factors that moderate the psychological consequences of the covid pandemic. I have some issues that I’d like to see addressed.

Please clarify if this was a study specifically based on older people, or if this was a general population study and data from older people have been extracted. If it was the later then some other data could be used for comparison purposes.

“520 older Israeli adults aged 67 and older”. Why 67, this seems an arbitrary age. Were inclusion/exclusion criteria used in the recruitment process?

Please ensure that all measures are described in sufficient detail (total # items in scale and subscales, Likert format and category numbers, possible range of scores for scale and subscales, indicate what high scores reflect, reliability estimates, any norms/cut-off scores that can be used to help interpretation of descriptive statistics).

Please add effect sizes to the t-tests and provide guidance on their interpretation.

---

## [Reviewer Report]

The investigation of factors associated with ICD-11 PTSD and CPTSD in the elderly is undoubtedly relevant. However, I have some doubts about whether the manuscript can be published without some major revisions. My comments:

• My main concern about the manuscript is related to the power of statistical analysis. Even though the total sample size is good, the groups of interest are very small (2.5% (n=13) of the sample each). The data collected is valuable, but the analyses raise the issue of the trustworthiness of the results. Maybe the analysis strategy could be reconsidered. Another option would be to explore statistical methods that could help to take into account the issue of a very low prevalence of a condition of interest.

• It‘s a bit hard to grasp how COVID-19-specific this study really is. The authors try to emphasize the role of the pandemic on the symptoms of PTSD/CPTSD. But what is really pandemic-specific here in this study? We already know that more exposure to traumas is linked with posttraumatic reactions; or loneliness - with CPTSD (being cut off from relationships is actually one of the symptoms). The only part more clearly related to the pandemic is COVID-19 worries. I don‘t want to underestimate the importance of authors‘ work. However, I missed more strong justification for how this study gives us specific pandemic-related implications. Is the fact that the study had been conducted during the pandemic enough? Or maybe this study is not as pandemic-specific as emphasized in the manuscript?

• The instruments used in the study are described very briefly. More information is needed to understand what and how the study constructs have been evaluated. Now if one is not very familiar with the scales in advance, it raises difficulties in understanding the results. Furthermore, even though the ITQ is described in more detail, I was left a bit unsure about the algorithm used. Based on the original ITQ, endorsement of a symptom is defined as a score =2 or > 2 (from „moderately“ to „extremely“). In the manuscript, the authors state, that <... clinical levels were determined when ... high/very high level of suffering...>. Does this mean a different algorithm was used? All these symptom endorsement procedures should be described in more detail and clearly.

• Also, I would suggest being more clear in the manuscript that ITQ is not a diagnostic tool. It only can be used for screening for probable PTSD/CPTSD.

There are some more comments, but I won‘t go into detail as my suggestion is to revise the chosen analysis plan.

---

## [Reviewer Report]

July 13, 2023

Professor Judith Bass

Editor-in-Chief 

Cambridge Prisms: Global Mental Health 

Dear Professor Bass,

We herewith resubmit our paper entitled “Factors associated with ICD-11 posttraumatic stress disorder (PTSD) and complex PTSD among older adults during the COVID-19 pandemic" for consideration for publication in your journal, Global Mental Health.

We thank the you, the handling editor Prof. Nino Makhashvili and the expert Reviewers for helpful comments to assisted us in improving our manuscript.

We declare that the contents of this manuscript have not been copyrighted or published previously and are not under consideration for publication elsewhere.

The contents of this manuscript will not be copyrighted, submitted or published elsewhere while acceptance by the journal is under consideration.

We hope that our manuscript is now acceptable for publication. 

Sincerely,

Lee Greenblatt-Kimron, Menachem Ben-Ezra, Maayan Shacham, Yaira Hamama-Raz & Yuval Palgi

Corresponding Author: 

Lee Greenblatt-Kimron 

School of Social Work, Ariel University, Ariel, Israel University.

E-mail: leegr@ariel.ac.il

---

## [Reviewer Report]

The authors have responded to all my comments and provided adequate explanations that alleviated my previous concerns. I have no additional comments.